# Present and Future Role of Immune Targets in Acute Myeloid Leukemia

**DOI:** 10.3390/cancers15010253

**Published:** 2022-12-30

**Authors:** Daniela Damiani, Mario Tiribelli

**Affiliations:** 1Division of Hematology and Stem Cell Transplantation, Udine Hospital, 33100 Udine, Italy; 2Department of Medicine (DAME), University of Udine, 33100 Udine, Italy

**Keywords:** acute myeloid leukemia, drug resistance, immune escape, immune therapy

## Abstract

**Simple Summary:**

Despite increasing knowledge of the biological mechanisms leading to neoplastic transformation, the identification of various molecular targets for innovative drugs, refinements in supportive care, and the wider use of allogeneic transplants, the prognosis of acute myeloid leukemia (AML) remains dismal. In the last years, the role of the bone marrow microenvironment and of the immune system in favoring leukemic cell persistence has emerged both as a cause of treatment failure but also as a potential setting for novel therapies. The aim of the present paper is to review the biological evidence and clinical trials of immune-based treatments for AML.

**Abstract:**

It is now well known that the bone marrow (BM) cell niche contributes to leukemogenesis, but emerging data support the role of the complex crosstalk between AML cells and the BM microenvironment to induce a permissive immune setting that protects leukemic stem cells (LSCs) from therapy-induced death, thus favoring disease persistence and eventual relapse. The identification of potential immune targets on AML cells and the modulation of the BM environment could lead to enhanced anti-leukemic effects of drugs, immune system reactivation, and the restoration of AML surveillance. Potential targets and effectors of this immune-based therapy could be monoclonal antibodies directed against LSC antigens such as CD33, CD123, and CLL-1 (either as direct targets or via several bispecific T-cell engagers), immune checkpoint inhibitors acting on different co-inhibitory axes (alone or in combination with conventional AML drugs), and novel cellular therapies such as chimeric antigen receptor (CAR) T-cells designed against AML-specific antigens. Though dozens of clinical trials, mostly in phases I and II, are ongoing worldwide, results have still been negatively affected by difficulties in the identification of the optimal targets on LSCs.

## 1. Introduction

Acute myeloid leukemia (AML) is a heterogeneous group of hematopoietic clonal diseases characterized by the accumulation in the bone marrow (BM) and in the peripheral blood (PB) of immature myeloid cells with increased proliferation capacity and survival ability, resulting in bone marrow failure and increased hemorrhagic and infective risk [1]. AML is the most common type of acute leukemia in adults, its incidence increases with age, and despite a consistent proportion of patients initially responding to therapy, the long-term prognosis remains quite poor, as over 60% of patients will eventually relapse and die from the disease [2]. Over the past twenty years, the role of clonal evolution in the leukemogenic process has emerged, identifying many recurrent molecular abnormalities, refining risk across the cytogenetic alterations, and, at the same time, identifying potential therapeutic targets [3,4]. The progress of deciphering the molecular complexity of acute leukemia, the availability of target therapies, and the wider use of allogeneic hematopoietic cell transplantation (HCT) promise better responses in selected AML subsets [5], but the recent data of the American Cancer Society still estimate a 5-year overall survival (OS) rate of 30%, and the prognosis is even worse in elderly patients, with a 1-year expected OS of less than 10–15% [6]. The high relapse rate, even after HCT, is largely attributed to the persistence of leukemic stem cells, sharing or hijacking the BM microenvironment of normal hematopoietic progenitors. It is well recognized that stem cell niche contributes to leukemogenesis, but accumulating evidence suggests that the complex crosstalk with leukemic cells contributes to creating a permissive, protective immune environment able to affect responses to therapy and facilitate relapse [7]. On this basis, efforts have recently been focusing on the identification of potential immune targets on leukemic cells and on tuning novel immune-based techniques to adjust the niche and reactivate the immune system, thus restoring anti-leukemia surveillance. 

In this paper, we summarize the current data on the biological basis and clinical use of molecular or cellular therapies targeting leukemia cells to overcome drug resistance and to contrast immune evasion, potentially restoring the anti-leukemia immune surveillance.

## 2. Leukemic Cells in Their Bone Marrow Microenvironment: How They Induce Drug Resistance and Escape Immune Response

A hierarchical organization of bone marrow leukemic cells resembling the normal hematopoietic system, with a small subset of leukemic stem cells (LSCs) on the top of the pyramid, continually replenishing the more mature bulky population, was first described 25 years ago by Bonnet and coll. [8]. As for the normal counterpart, leukemic stem cells are defined by their self-renewal capacity and their ability to initiate leukemia after serial transplantation in SCID mice [8,9]. Interestingly, LSCs seem to originate from the acquisition of driver mutations, not by normal immature hematopoietic stem cells (HSCs) but by committed progenitors. At least two distinct LSC populations have been identified in human AMLs: a more mature population, emerging from granulocyte-macrophage progenitors (GMPs), and a more immature population, from lymphoid-primed multipotent progenitors (LMMPs) [10]. The recent identification of the clonal hematopoiesis of indeterminate potential (CHIP) [11] and the recognized ability of more mature LSCs to revert to an immature state [12] have further complicated the scenario. However, the persistence of pre-leukemic hematopoietic clonal alteration in responding patients confirms the negative role of LSCs [13]. Although it is well known that LSCs are enriched within the CD34+CD38- fraction, a clear definition of the LSC phenotype does not exist. Compared to normal HSCs, LSCs may have higher expression of CD25, CD32, CD44, CD96, CD123, CD200, GPR56, N-cadherin, Tie2, TIM-3, CLL-1, c-MPL, and HDM2 [14,15,16,17,18,19,20,21,22,23,24]. Unfortunately, the high inter- and intra-individual heterogeneity complicates their enumeration and sequential monitoring. Despite this, the negative prognostic role of LSC frequency at diagnosis or after therapy has been demonstrated by many studies, and different LSC prognostic scores have been obtained by gene expression profiles and whole exome plantation [25,26,27]. LSCs escape the effect of anti-leukemic drugs by nesting in the BM microenvironment in a quiescent state, and leukemic progeny gradually occupy the BM niche, converting it into a “leukemic niche”, able to support leukemic cells’ survival and proliferation while decreasing its capacity to maintain normal hematopoiesis [28,29,30]. It is interesting to underline that BM failure is not due to a reduction in HSCs but to its inability to produce an adequate number of progenitors as a consequence of the release by mesenchymal cells (MSCs) of hypoxia-associated molecules that increase stemness and prevent differentiation [31], mimicking the differentiation arrest that is the hallmark of the leukemia phenotype. The modifications of the BM niche by leukemic cells involve all the components of the microenvironment. The endosteal niche is reshaped by the loss of balance between bone formation and resorption as the consequence of the activation of the RANK/RANKL pathway, which promotes osteoclastogenesis and favors osteoclast survival [32]. Moreover, the secretion by LSCs of a bone morphogenic protein (BMP), of a negative regulator of osteogenesis (DKK), and of a chemokine that decreases osteocalcin and switches MSC differentiation from adipogenic to osteoblastic with the consequent accumulation of progenitor and immature osteoblasts and defects in bone mineralization (CCL3, also known as MIP1α) [33,34], contribute to creating a milieu of facilitating leukemia cell growth and AML progression. Furthermore, in vitro data suggest that MSCs also support leukemia cell survival and promote resistance to chemotherapy, inducing a low cycling rate and anti-apoptotic signals [35,36]. 

In the AML vascular niche, increased IL-1β and TNF-α levels synergize with increased adhesive receptors E-selectin/CD44 and VCAM-1/VLA-4, thus ensuring the anchorage of LSCs to endothelial cells [37,38]. VEGF/VEGFR and Notch/Delta-like ligand 4 promote neo-angiogenesis, essential for AML progression and extramedullary homing [39,40]. Although less known compared to the other, the reticular niche is considered a transitional niche, able to maintain HSCs in a proliferative state, such as the vascular, but in an undifferentiated state, such as the endosteal one. CXCL12-abundant reticular (CAR) cells and Nestin-expressing cells are the most represented cells in the niche, localized in proximity to sinusoidal endothelial cells [41,42,43]. Most HSCs are in contact with CAR cells, but interaction with these cells is established also by B-lymphocytes, plasma cells, plasmacytoid dendritic cells, and NK cells, suggesting that reticular cells might also act as an immune cells niche [44,45,46]. CAR cells secrete CXCL12 and SCF, crucial for HSCs trafficking and homing. In AML, the CXCL12/CXCR4 axis regulates the infiltration of leukemic cells in the protective bone marrow niche [47,48]. Nestin-expressing cells are associated with adrenergic nerves and regulate HSC maintenance [43,49]. AML bone marrow is enriched with Nestin-expressing cells. Besides their role in developing AML, Nestin-expressing cells are implicated in the induction of resistance to chemotherapy by enhanced glutathione (GSH)-peroxidase (Gpx) activity [50].

Considered in the past only as passive bystanders, BM adipocytes actually play an important role in regulating normal HSCs and are implicated in hematopoietic recovery after irradiation or 5FU treatment by secreting SCF [51]. In AML, they promote leukemic cell survival during therapy and drug resistance by enhancing free fatty acid production [52,53] and by sequestering, inactivating, and metabolizing chemotherapeutic drugs [54]. Among cells from myeloid resident in the bone marrow, myeloid-derived suppressor cells (MDSCs) and leukemia-associated macrophages (LAMs) are also implicated in generating an immunosuppressive microenvironment in AML. MDSCs are a heterogeneous population, able to induce T-cell tolerance by multiple mechanisms, still not completely elucidated. It is well known that MDSCs express high levels of V-domain Ig suppressor of T-cell activation (VISTA) and PD-L1, but also indoleamine 2,3-dyoxigenase (IDO), arginase, ROS, TGFβ, and IL-10 [55,56,57,58]. In AML, there is an increase in MDSCs by the release of extracellular vesicles containing the oncoprotein MUC1, which induces MDSC expansion via c-myc [59]. Some clinical data suggest that MDSC numbers correlate with disease progression and poor survival [60,61]. MDSCs are distributed through the three niches, in proximity to leukemic cells. As for MDSCs, resident BM macrophages are not localized in a specific niche, but they are able to adapt to the microenvironment, where they play a role in maintaining homeostasis and in processing and presenting antigens to adaptive T-cell effectors. In the AML microenvironment, they may be involved in the protection of leukemic cells from apoptosis induced by chemotherapy [62]. The increasing knowledge of the healthy BM microenvironment and of its dynamic remodeling to become a leukemic “sanctuary” has completely changed the old view of leukemogenesis as merely a product of accumulating genetic defects, highlighting its dependence on active crosstalk with BM niches. This perspective paves the way toward new, challenging treatment strategies, including targeting adhesion or leukemic cell/niche cell signaling pathways, reversing bone remodeling or signals, inducing vascular disruption, and metabolic changes counteracting drug resistance. 

Several clinical trials are currently ongoing to investigate the efficacy and toxicity of many niche-targeting molecules. Their description is beyond the scope of this paper, but an exhaustive overview of this topic was recently published by Kuek and coll. [63]. Besides the stromal component of the niche, an important role in creating a permissive environment is played by the immune cells resident in the BM. Although spatially distinct T and B compartments are not recognizable, it is well known that a significant proportion (8–20%) of innate and adaptive immune cells transit and reside in the BM with a T cell/B cell ratio of 5:1 [64,65]. Marrow lymphocytes are distributed throughout stroma and parenchyma and condensed in follicles; antibody-producing plasma cells account for about 1% of mononuclear cells. Despite most T cells having already encountered an antigen, making the BM an “immune memory reservoir”, some circulating naive T cells can be primed in the BM [65]. In healthy individuals, T cells benefit from the BM environment and, at the same time, support hematopoiesis through cytokine secretion and the expression of chemokine receptors [66]. BM T cells have an active role in allogeneic stem cell engraftment and memory. CD8 T cells exert an anti-leukemia effect by limiting the potentially harmful graft versus host disease (GVHD) [67,68,69]. Moreover, they are capable of anti-viral activity, providing protection from chronic infections [70]. As well as BM effector T cells, also T regulatory cells (Tregs), which are present in the BM in higher numbers than in peripheral blood and lymphoid organs, contribute to hematopoiesis maintenance by regulating the endosteal niche [66]. Natural killer (NK) cells play a protective role against GVHD in transplant settings and have a potent anti-tumor effect [71,72]. BM is also the primary B-lymphoid organ, where B precursors develop from HSCs and mature in specific niches rich in CXCL12- and IL7-expressing cells [44]. Furthermore, the BM acts as a reservoir for long-lived plasma cells, providing survival factors, and therefore contributing to maintaining long-term immunity [73]. Beyond remodeling the stromal niche, marrow AML cells implement several unique strategies to escape immune surveillance by directly affecting the function of immune response players and preventing immune-mediated elimination. AML cells can develop immune-editing processes to reduce cell recognition, leading to the epigenetic downregulation of HLA molecules, the loss of HLA molecules, the increased production of immunosuppressive/inhibitory molecules, the increased expression of inhibitory ligands, the expression of checkpoint inhibitors, the altered expression of NK cell ligands, and the impaired capacity of T cells to form immune synapsis [58,74,75]. Moreover, they can induce the expansion of Tregs, MDSCs, and the M2 polarization of macrophages [75]. A schematic representation of the immune evasion mechanisms of AML cells is shown in Figure 1.

A longitudinal investigation of the immune landscape in leukemia patients had not only demonstrated a high inter-patient variability but revealed also that a selective immune signature predicts a negative outcome. Tang and coll. reported a negative event-free survival (EFS) and OS in the presence of CD8+PD-1+ T cells [76]. Guo and coll., mapping a series of AML-derived BM immune cells, confirmed the diverse immune assets among patients. They also identified many infrequent immune cell types, such as TH17/Treg intermediate population, CD8+ memory T cells, dysfunctional macrophages, and dendritic cell subsets, predicting a poor prognosis [77]. Rutella and coll. investigated AML-driven CD8+ exhaustion/senescence, developing an immune effector dysfunction (IED) score associated with leukemia stemness and poor response to therapy and possibly predicting resistance to immunotherapy [78]. On this basis and following the success in lymphoid malignancies and in multiple myeloma, many immune therapies for AML are under development. The major challenge is the identification of the target, given the high heterogeneity of the AML population, the difficulty in finding antigens restricted to LSCs, the lower antigen density compared to lymphoid cells, as well the best therapeutic strategy (in combination with chemotherapy and/or molecular therapy or sequentially), and the choice between an antibody-based or cellular-based therapy.

In the last few decades, different strategies have been developed to increase therapeutic potential: targeting LSC-restricted antigens, reactivating endogenous T-cell response through immune checkpoint inhibitors, and harnessing T-cell and NK response independently of T-cell receptor (TCR) specificity through cell-engaging antibody constructs and the genetic engineering of T cells (TCR-modified and chimeric antigen receptor (CAR) T cells). 

## 3. Monoclonal Antibodies

### 3.1. Potential Targets on LSCs

The main mechanism of action of unconjugated antibodies is antibody-dependent cell-mediated cytotoxicity (ADCC). After the formation of an immunological synapse with the target cell, NK cells trigger a cytolytic response through the exocytosis of granules containing perforin and granzyme into the target cell. Furthermore, they facilitate antibody-dependent phagocytosis (ADCP) by macrophages. Their major limitation is the presence of inhibitory signals counteracting their action and the limited potency of activation signals.

#### 3.1.1. CD33

CD33 is a 67 kDa glycoprotein, a member of the siglec family (siglec-3), expressed in normal hematopoiesis from early myelo-monocytic lineage-committed progenitors to mature cells. It is also expressed in 99% of AML cells and in LSCs [79]. The use of unconjugated anti-CD33 is challenging due to its low membrane density and the low antibody-induced internalization, such that the first phase 3 trial (NCT0006045) with humanized IgG1 unconjugated antibody anti-CD-33, lintuzumab (HuM195) was stopped due to lack of efficacy [80]. Despite this, four phase I/II trials are ongoing to test a new anti-CD33 unconjugated IgG1 (BI 836858), alone or in combination with azacytidine (AZA), decitabine (DAC), or F16-IL2, in relapsed/refractory and newly diagnosed AMLs (NCT01690624, NCT03013998, NCT02032721, NCT03207191). The principal advantage of antibody–drug conjugates is the simultaneous selective attack of target cells and the on-site delivery of the potent conjugated cytotoxic agent, not usable as free drugs in conventional chemotherapy schemes for their toxicity. After the spontaneous withdrawal of the manufacturer due to the lack of survival improvement and the high induction mortality, the anti-CD33-calicamycin conjugate gemtuzumab-ozogamicin (GO) has been revived for core binding factor (CBF) AMLs and for elderly patients with AML or high-risk myelodysplastic syndrome (MDS), where it has demonstrated reduced relapse risk and survival advantage in patients with favorable and intermediate cytogenetic risk [81]. A phase I clinical trial testing the safety and efficacy of GO in combination with venetoclax (VEN) in adult relapsed/refractory AMLs is currently ongoing (NCT04070768). A new anti-CD33 conjugate with DGN 462, a DNA-alkylating drug, was active in a preclinical model [82], and a phase I trial in CD33+ relapsed/refractory AMLs has been terminated but the results are not yet available (NCT02674763). 

#### 3.1.2. CD123

CD123 is the interleukin (IL)-3 receptor alpha chain and is a type I transmembrane glycoprotein [83]. CD123 is present in 98% of CD34+CD38- LSCs and on blast cells but not in normal hematopoietic progenitors, making it a potential therapeutic target [17]. The unconjugated antibody (talacotuzumab) failed to demonstrate efficacy in AML and MDS (NCT0299860). The phase I/II trial with anti-CD123-DGN462 conjugate, alone or with AZA± VEN in MRD+ post-induction therapy AMLs, is still recruiting patients (NCT04086264). 

#### 3.1.3. CLL-1

C-type lectin-receptor1 (CLL-1) is an ITIM-containing inhibitory transmembrane protein code on chromosome 12p13.31. CLL-1 is not expressed in normal HSCs, but it is present in committed progenitors and in mature peripheral myeloid cells such as monocytes, granulocytes, and dendritic cells [84]. Its function, as well as its ligand, are not completely understood. It has been suggested to have involvement in regulating some inflammatory situations by down-modulating granulocyte and monocyte functions [85]. Being selectively expressed in LSCs, even more than CD123, which is also present in some CD34+CD38- normal progenitors, CLL-1 could be regarded as an ideal target for anti-LSC immune therapies [22,86]. Despite CLL-1 efficiently internalizing after ligand binding, in vitro experiments with an anti-CLL-1 antibody suggested that an un-conjugated antibody cannot have anti-leukemic activity because it does not activate ADCC [22]. In animal models, two antibody–drug conjugates with pyrrolobenzodiazepine (DCCL9718A) and isoquinolidinobenzodiazepine (CLT030) induced a potent anti-leukemic effect, with little off-target toxicity [87,88]. To enhance anti-CLL1 efficacy, different bispecific antibodies and cell-based therapies have been developed (see below).

#### 3.1.4. Other Current Clinical Trials of Toxin-Conjugate Antibodies

FLT3 (FMS-like tyrosine kinase 3) is a member of the class III receptor tyrosine kinase family that is highly expressed in the blasts of both AML and ALL patients. FLT3 plays an important but not absolute role in maintaining the survival of normal HSCs, and its recurrent mutations (ITD-FLT3, TDK-FLT3) are expressed in many AML cases [89,90]. Working in conjunction with other growth factors, FLT3 promotes the proliferation and differentiation of myeloid and lymphoid cells. Transplant animal models with non-functioning and wild-type FLT3 showed that hematopoiesis is almost normal in FLT3 knockout animals, while FLT3 mutations give a significant growth advantage. These facts suggest that selective FLT3 inhibition in leukemia cells can block excessive FLT3 leukemia activation with acceptable hematopoietic side effects [90]. The phase I trial (NCT02864290) testing AGS62P1 anti-FLT3 antibody-amberstatin 269 in relapsed/refractory adult AMLs has been closed for lack of efficacy. NCT03957915, a phase I trial with INA03 drug-conjugate antibody-targeting transferrin receptor (CD71), is active, not recruiting, and NCT01830777, testing brentuximab-vedotin in CD30+ relapsed AMLs, is completed (results not yet available). More recently, the use of monoclonal antibodies labeled with radionuclides has been proposed, especially in the setting of HCT conditioning regimens, to obtain a more precise delivery of ionizing radiation to the disease site [91]. This method would maintain the advantages of high-dose total body irradiation (TBI) and the low toxicity of reduced-intensity regimens. At present, many trials with radiolabeled antibodies are ongoing. In the phase III SIERRA trial (NCT02665065), patients are randomized to receive conventional myeloablative versus ^131-^I-BC8, an iodine-conjugate anti-CD45 antibody. The preliminary results on the first 25% of patients, presented at the ASH Meeting in 2018, demonstrated the protocol feasibility, though non-relapse mortality was relevant in the ^131-^I BC8 arm, and a clear reduction in toxicity was not evident. The still-recruiting NCT03867682 trial investigates the maximum tolerated dose and remission state in adult relapsed/refractory AML receiving a radiolabeled anti-CD33 conjugate (^225-^ Ac-Lintuzumab) + VEN. The phase I trial NCT 03,441,048 is recruiting patients with R/R AML to evaluate the toxicities of the combination of a CLAG-M regimen with ^211-^At-BC8, an anti-CD45 antibody. NCT 03670966, a phase I/II trial, investigates the toxicity, GVHD incidence, donor chimerism, engraftment rate, and 100-day survival of the combination of ^211-^At-BC8 with a reduced-intensity preparative regimen (fludarabine, cyclophosphamide, 2-Gy TBI) in haploidentical stem cell transplantation in patients with R/R AML. The same endpoints are investigated in the phase I/II NCT03128034 trial, in which 211At-BC8 is employed with fludarabine and 2–3 Gy TBI. The information from all these trials should allow a further reduction in toxicity through individualized treatment in line with the emerging “theranostics” concepts. 

### 3.2. Immune Checkpoint Inhibitors

The immune system activity is closely modulated by the interaction between co-inhibitory molecules and their ligands. These co-inhibitory molecules are involved in the maintenance of immune tolerance, but in the neoplastic setting may represent one of the mechanisms employed by cancer cells to escape immune surveillance [92,93]. Many studies have reported an increased expression of co-inhibitory molecules in solid cancers, and, more recently, similar results have been observed in AML patients [94,95]. The up-regulation of co-inhibitory ligands has been associated with poor clinical outcomes in solid and hematological cancers [93,94], thus identifying a potential new class of therapy targets (Figure 2). Many molecules, such as ipilimumab, pembrolizumab, cemiplimab, avelumab, and durvalumab, with different blockade targets, have been approved for the treatment of solid tumors, and are under investigation in AML [96]. Single-agent treatments proved the ability of co-inhibitory blockade to improve immune response in many tumors, but at the same time highlighted the modest results of a monotherapy, promoting the design of various combination trials. About fifty trials, designed to provide definitive information on safety and long-term efficacy, are currently recruiting patients with AML, testing molecules blocking different co-inhibitory axes and different drug combinations.

#### 3.2.1. The CTLA-4/B7 Axis

CTLA-4 (CD152) is a member of the immunoglobulin-related receptors interacting with CD80 and CD86 ligands to deliver an inhibitory signal to terminate immune responses. Furthermore, it is involved in the induction of Tregs lymphocytes, playing a pivotal role in regulating tolerance and autoimmunity [97]. Aberrant expression has been reported in AML, often in association with the expression of other checkpoint inhibitors, and with a negative impact on disease outcome [98]. In murine tumor models, the blocking of CTLA-4 enhanced T-cell activity, suppressing Treg cells [99]. In clinical trials for melanoma, the anti-CTLA-4 agent ipilimumab increased the Teff/Tregs ratio, enhanced NK activity, and restored T effector function, ultimately prolonging survival [100,101]. In vitro tests on AML cell lines and preliminary clinical studies confirmed the data on solid tumors [102]. The available “in vivo” data on ipilimumab efficacy is scarce. Davids and coll. reported 23% complete remission (CR) and 9% partial response (PR) in a series of hematological malignancies treated with single-agent ipilimumab for relapse after HCT [103]. At present, a phase I trial exploring the safety and toxicity of ipilimumab combined with a definite dose of Treg-depleted donor lymphocyte infusion (DLI) for AML and MDS relapse after transplant is recruiting (NCT03912064). 

#### 3.2.2. The PD1/PD-L1 Axis

PD1 (CD279) is a type I transmembrane protein preferentially expressed in almost all activated immune cells [92]. PD1 binds two different ligands: PD-L1 (CD274) and PD-L2 (CD273). PD-L1 is a member of the B7 family of co-stimulatory/co-inhibitory molecules also present in hematopoietic cells and upregulated or aberrantly expressed in many tumors [92,104]. PD-L1 and PD-L2 over-expression in tumor cells has been associated with poor outcomes [94]. PD1/PD-L1 engagement produces an inhibitory signal that causes T-cell exhaustion and favors tumor cell immune escape [92]. Moreover, it induces the apoptosis of tumor-specific cells and favors Tregs differentiation and resistance to CD8+-mediated cytolysis [105]. In AML, PD1 up-regulation was described in T effector cells and Tregs, explaining the immune suppression observed during AML progression. PD-L1 and PD-L2 were found in blast cells and correlate with poor prognosis [106,107]. Despite the proven activity in solid tumors, PD1 and PD-L1 inhibitors are, to date, not approved for AML. The preliminary results of a phase II study of nivolumab maintenance in high-risk AML patients in CR after induction and consolidation (NCT02532231), presented by Kadia and coll. in 2018 at the Annual Meeting of the American Society of Clinical Oncology (ASCO), report CR rates of 79% and 71% [108]. Many clinical trials combining epigenetic therapies with nivolumab or durvalumab in AML patients are recruiting. The results of combination therapy with AZA and nivolumab in relapsed AML (NCT02397720) presented in 2017 at the ASCO Annual Meeting reported 21% CR/Cri and a hematologic improvement in 7%, with a median OS of 9.2 months [109]. Finally, grade 2–4 adverse immune events were observed in 28% of patients. In another phase II clinical study associating nivolumab with cytarabine and idarubicin in newly diagnosed AML and MDS (NCT02397720), 72% achieved CR/CRi and 4-week and 8-week mortality were stable at 6% [110].

#### 3.2.3. Tim-3/Galectin-9 Axis

T-cell immunoglobulin and mucin domain 3 (TIM-3) is a cell-surface molecule first identified on CD4+ TH1 cells and on CD8+ cytotoxic T cells (CTLs) and later on other innate immune cells, such as dendritic cells, monocytes, macrophages, mast cells, NK cells, and in many neoplastic cells [111,112,113]. The TIM-3 gene is coded on chromosome 5q33.2, in the same region of IL4 and IL-5 genes. There are four known ligands for TIM-3: the first recognized and most widely studied is galectin-9 (gal-9), which induces the apoptosis of TH1 cells [114], playing a crucial role in tumor cell immune evasion. Its high expression has been associated with a worse prognosis in solid and hematologic neoplasms [92,94]. In AML, high levels of TIM-3 have been described in immune cells, associated with immune exhaustion, and on LSCs, where it represents a distinctive marker. Kikushige and coll. suggested that on LSCs, TIM-3 and its ligand create an autocrine loop that regulates development and self-survival [21]. Taken together, these data identify TIM-3 as an excellent candidate for therapy with monoclonal antibodies. Currently, several monoclonal antibodies are being used in clinical trials for solid tumors, such as MBG453 (sabatolimab), TSR-022, BMS-986258, LY3321367, SYM023, BGB-A425, and SHR 1702 [115], but only MBG453 has shown preliminary efficacy and safety in AML and in MDS. Most of the trials are still ongoing. The interim data of trial NCT 03,066,648 (MBG453 in combination with HMAs) showed an ORR of 58% in MDS and 48% in newly diagnosed AML patients for the MBG453 + DAC arm and 70% in MDS and 27% in AML patients for the MBG453 + AZA arm, respectively. The most common grade 3/4 adverse events are thrombocytopenia, anemia, neutropenia, and febrile neutropenia; in the MBG453 + DAC group, four immune-related events were reported (ALT, arthritis, hepatitis, hypothyroidism), compared to none in the MBG453 + AZA cohort. 

#### 3.2.4. LAG-3/MHC Axis

LAG-3 (CD223) is a CD4-like molecule that binds MHC class II with higher affinity compared to CCD4, generating a signal that blocks T-cell activation [100]. The molecule is expressed on activated T cells, Tregs, NK cells, B cells, and dendritic cells [94]. The engagement with its ligand reduces T-cell activity and cytokine secretion, blocking T-cell activation and function [100]. Like other checkpoint molecules, LAG-3 has been identified on Tregs in the cancer microenvironment. The frequent co-expression with PD1 suggests a comparable function to PD1 [101,105]. Studies in AML are still limited [116]. It should be remembered that the expression of MHC class II in AML cells can be involved in both immune suppression and antigen presentation processes. Antibody-targeting LAG-3 is currently being tested in solid tumors, lymphoma, and multiple myeloma [94,100]. No clinical trials have been initiated in AML patients.

#### 3.2.5. CD200/CD200R Axis

CD200 is a highly conserved 48kDa type-1 transmembrane glycoprotein, structurally related to the B7 family, coded on chromosome 3q12-q13 near the region coding for CD80/CD86 proteins. Interaction with its ligand CD200R leads to the attenuation of many immune-responsive effects, resulting in the prolonged survival of transplanted allographs but also tolerance for tumor cells. CD200 overexpression has been described in several solid tumors and in AML, where it marks LSCs but not the normal HSC counterpart [18,94]. CD200 overexpression in AML has been associated with poor outcomes, especially in the favorable prognostic groups [117,118]. Furthermore, the expression of CD200 on leukemic cells suppresses the function of memory T cells, expands Tregs, and down-modulates NK function [119,120,121]. “In vitro” and “in vivo” mouse models have clearly demonstrated that the inhibition of the CD200/CD200R axis with monoclonal antibodies restores the anti-leukemia immune response [122]. At present, the monoclonal antibody anti-CD200 samalizumab is under investigation in solid tumors, but there are no active trials in AML.

#### 3.2.6. CD47/SIRPs Axis

Leukocyte surface antigen CD47, encoded on chromosome 3q13.12, is a transmembrane glycoprotein that serves as the ligand for signal regulatory protein alpha (SIRPα), which is expressed on phagocytic cells, including macrophages and dendritic cells. Upon activation, CD47 initiates a signal transduction cascade, resulting in the inhibition of phagocytosis [123]. CD47 is broadly expressed in a variety of normal tissues, in normal circulating HSCs, as well as leukemic cells [124]. Moreover, its expression in leukemia cells has an adverse prognostic effect [125]. Preclinical studies demonstrated that a block of CD47 exerts an anti-neoplastic effect on solid tumors and hematologic malignancies [126,127].

Magrolimab is the first-in-class monoclonal antibody against CD47 available for clinical use. In 2020, the FDA granted breakthrough therapy designation to magrolimab based on positive results in MDS. In high-risk MDS, the phase 1b trial of combination magrolimab + AZA demonstrated a favorable safety profile: the most common treatment-emergent adverse events (TEAEs) included constipation (68%), thrombocytopenia (55%), anemia (52%), neutropenia (47%), nausea (46%), and diarrhea (44%). CR and ORR were 33% and 76%, respectively. OS rates at 12 and 24 months were 75% and 52%, respectively. Favorable outcomes were observed both in patients with TP53 mutation (40%) and WT TP53 (31%). A phase 3 trial of magrolimab/placebo + AZA in MDS is recruiting (ENHANCE trial—NCT04313881). A similar tolerability and safety profile was observed in a phase 1b trial investigating the association of magrolimab + AZA in frontline patients with TP53-mutated AML. Common TEAEs were constipation (52.8%), diarrhea (47.2%), febrile neutropenia (45.8%), nausea (43.1%), fatigue (37.5%), decreased appetite (37.5%), thrombocytopenia (31.9%), peripheral edema (30.6%), and cough (30.6%). ORR was 48%, with CR 33.3% and CRi 8.3%. Thirty- and sixty-day mortality was 8.3% and 18.1%, respectively. The median OS was 10.8 months. In high-risk AML patients with mutated TP53 unsuitable for intensive therapy, magrolimab + AZA showed a durable response and encouraging OS. A phase 3 trial of this combination vs. standard care in TP-mutated AML is ongoing (ENHANCE2, NCT04778397). Another phase 2 clinical trial of magrolimab + AZA + VEN vs. magrolimab + intensive chemotherapy (mitoxantrone, etoposide, and cytarabine—MEC) vs. magrolimab + oral AZA (CC-486) is open and recruiting (NCT04778410). At the 2022 Annual ASH meeting, Daver and coll. presented the results of a phase IIb/II trial with the triplet combination therapy AZA–VEN–magrolimab in 74 elderly patients with newly diagnosed, ELN high-risk AML (NCT04435691), reporting an ORR of 74%, CR/CRi 63%, and a 4–8-week mortality of 0%. Anemia was the major hematological adverse event; the most common non-hematological AEs were febrile neutropenia (50%), pneumonia (38%), hyperbilirubinemia, and transaminitis (11%). They concluded that the combination AZA–VEN–Magro was safe, with encouraging CR rates, and initiated a phase III placebo-controlled, randomized study to evaluate this triplet in ND AML (ENHANCE-3, NCT05079230) [128].

Evorpacept (ALX148) is an engineered high-affinity CD47-blocking protein, in which the Fc domain has been modified to avoid the red cell agglutination experienced with magrolimab. There are two ongoing trials assessing ALX148 in combination with AZA in high-risk MDS (ASPEN02 trial—NCT04417517) and in combination with AZA + VEN in AML (ASPEN05 trial—NCT04755244); the results of the phase 1 part of the trial have been recently published, showing good tolerability of the combination of evorpacept + AZA [129]. TEAEs observed in >1 subject included constipation and infusion-related reactions (23%) and nausea or vomiting (15%); there were no evorpacept-related SAEs, and no patients discontinued treatment due to an AE. The combination will be further evaluated in the randomized phase 2 part of the study.

### 3.3. T Cells Engagers

Given the unsatisfactory results of “naked” and drug-conjugate antibodies in treating AML, over the past years, combination strategies to overcome tumor cell and immune cell resistance have been developed. The purpose was to harness the immune response by targeting specific tumor antigens and engaging the dormant effector cells (T cells, NK-cells, macrophages) against leukemic cells to obtain a fast and potent cytotoxic response and, ideally, also generate a durable immunologic memory [130]. On this basis, several bispecific T-cell engager (BiTE) antibodies have been developed for hematologic malignancies, and novel structures are constantly emerging [131]. Bispecific antibodies consist of a single heavy and light chain of the variable region of a tumor-associated antigen and CD3: by combining the two different specificities, they can activate exhausted T cells through sustained tumor antigen exposure, overcoming their possible low-density expression. At the same time, as the T cells are activated only in the presence of target cells, they have limited off-target cytotoxicity. Moreover, BiTEs act in an MHC-1-independent manner, generating a cytolytic synapsis between CD8+ T cells despite the tumor-induced MHC-1 down-modulation, and permits T-cell activation even in the absence of co-stimulatory signals such as CD28 or IL-2 [132]. Starting from blinatumomab, the first CD19/CD3 BiTE approved by the FDA in 2014 for adult Ph+ acute lymphoblastic leukemia (ALL), many other BiTEs have been developed. Construct variants include dual affinity retargeting (DARTs) antibodies and bi- and tri-specific killer engagers (BiKEs and TriKEs). In their basic construct, BiTEs are connected by a linker molecule that defines the flexibility and the antigen-binding kinetics. In DARTs, a c-terminal disulfide bridge is included to improve stabilization, resulting in stronger T-cell activation and cell lysis [133]. BiKEs and TriKEs reactivate the immune system by engaging NK cells via CD16; upon stimulation, they release cytokines, leading to tumor cell lysis and recruiting other immune cells to amplify immune response [132]. Furthermore, TriKes have an IL-15 crosslinker, which drives NK expansion and increases killing response [134].

#### 3.3.1. Anti-CD33

The CD33 was integrated with an AMG330 bispecific antibody. Pre-clinical studies demonstrated potent T-cell activation and cytokine release, high blast cell clearance, and a reduction in BM monocytic MDSCs [135,136]. Moreover, AMG330 lytic activity was not affected by CD33 polymorphisms nor by the over-expression of drug transporter proteins on AML cells, as happens for gemtuzumab ozogamicin, and it does not modify CD33 membrane density, as conventional anti-CD33 antibody does [137]. On this basis, AMG was administered in a phase 1 dose-finding trial (NCT02520427) for patients with relapsed/refractory AML. A total of 55 patients were enrolled in 16 cohorts, receiving increasing doses. CR was obtained in 8/42 (19%) evaluable patients, despite half of the responding patients being heavily pretreated. Cytokine-release syndrome (CRS) and nausea were the most common adverse events (67% and 20%, respectively) [138].

AMG 673 is an anti-CD3-CD33 BiTE with a modified construct on the Fc region to prolong its half-life, permitting weekly infusions. In the NCT 03,224,819 phase I trial in R/R AML, it has demonstrated efficacy in reducing BM leukemia burden in 44% of patients, even if CR was achieved in only one patient. CRS was experienced by 50% of patients, transaminitis by 17%, leukopenia by 13% (febrile neutropenia by 7%), and thrombocytopenia by 7%.

AMV 564 CD3-CD33 BiTE was studied in animal models, demonstrating a potent anti-leukemic effect in BM and PB. In a phase 1 trial in R/R AML (NCT03144245), it confirmed anti-leukemic activity via T-cell activation, irrespective of antigen expression level. Moreover, AMV 564 was able to deplete MDSCs in a dose-dependent manner. Preliminary results on toxicity in 18 patients show an acceptable profile, with CRS grade 2 in one patient and febrile neutropenia in 7/18 patients (38%) [139].

G333 is a novel CD3-CD33 BiTE, modified with a linker to increase T cell binding to AML cells. In preclinical studies, it has demonstrated a low effect on normal HSCs [140], while a phase I trial in R/R AMLs is ongoing (NCT03516760), with results not yet available.

JNJ-67571244, CD3-CD33 BiTE, was tested in a phase I, dose-escalating trial in R/R AML and high-risk MDS (NCT03915379) that enrolled 68 patients; the trial was completed on March 28th, 2022, but the results have still not been reported.

GTB-3550 is a CD16-CD33 BiKE that, in preclinical studies, demonstrated the ability to overcome MHC-1 inhibitory signals, exert anti-leukemic activity, and reverse the MDSCs-induced suppression of NK cells. A phase I trial in R/R AML, MDS, and advanced systemic mastocytosis is currently ongoing (NCT03214666). No data are available currently.

#### 3.3.2. Anti-CD123

Vibecotamba (XmAb14045), is a potent CD3-CD123 bispecific antibody, showing in preclinical models high anti-leukemic activity on PB and BM blasts and a long half-life [141]. The initial results of a phase I dose-escalation study in R/R AML (NCT 02730312) enrolling 104 patients showed an ORR of 14%, a CR rate of 4%, and stable disease in 71% of patients. CRS was observed in 59% of patients. CD123 expression intensity on AML blasts seemed not to affect response [142].

APVO436, another CD3-CD123 bispecific antibody, is currently in a clinical trial on AML and MDS (NCT03647800). One of its peculiar features seems to be a low induction of cytokine release while maintaining the ability to reactivate T cells [143]. However, preliminary results from the ongoing trial suggest a limited efficacy [144].

Flotetuzumab (MDG006), a CD3-CD123 DART antibody, has been tested in a phase I/II clinical trial for R/R AML (NCT02152956), demonstrating anti-leukemic activity in 30 heavily pretreated patients, with an ORR and a median OS of 10.2 months among responders [145]. Interestingly, in vitro studies demonstrated an up-regulation of PD1/PD-L1 following flotetuzumab, prompting its combination with an anti-PD antibody, currently under investigation [146]. Further, MGD006 has demonstrated the preferential binding of leukemic cells and blast cell lysis at low doses, along with the maintenance of BM cellularity and HSC compartment [147].

Recently, the dual-targeting triple-body SPM-2, targeting CD33 and CD123 and engaging NKs as effector cells via CD16, has been tested on blast cells from 29 patients with AML. The new construct has demonstrated lytic activity across different AML subtypes at nanomolecular concentrations, raising the expectation that SPM-2 may also be capable of eradicating LSCs [148].

At the 2022 ASH Annual Meeting, Bergér and coll. presented the preclinical results of the combination of anti-CD123 and CD200 as prodrugs of an on-target activating trispecific T-cell engager. They demonstrated its ability to kill AML in vitro and in animal models, and its potential to eliminate AML LSCs while minimizing the risk for severe on-target off-leukemia toxicity [149].

#### 3.3.3. Other Targets

MCLA-117 is a CD3-CLEC12A (CLL-1) bispecific antibody currently under investigation in a phase I trial of R/R AML and newly diagnosed elderly AML patients (NCT03038230). In vitro studies demonstrated its ability to lyse cells at a low E:T ratio, even blast cells with low CLL-1 expression [150]. Its extended half-life permits weekly or bi-weekly infusions [151]. CLEC12a TriKEs have been developed to target LSCs lacking CD33 expression, but expressing CLL-1, demonstrating lower off-target toxicity and similar lytic capacity compared to the CD33 target, preserving normal HSCs [152].

AMG427 is an anti-CD3-FLT3 (CD135) bispecific antibody constructed to target the same receptor expressed from leukemic cells. It has demonstrated in vitro T-cell-dependent cytotoxicity but also the increased production of PD1 expression, which can act as an immune escape mechanism without the concurrent administration of anti-PD1 therapy [153].

At the 2022 ASH Annual Meeting, Sammicheli and coll. reported on the preclinical evaluation of ISB1442, a first-in-class CD38 and CD47 bispecific antibody for the treatment of AML and T-ALL. They demonstrated the ability of ISB1442 to induce the killing of primary tumor samples from AML and T-ALL patients in a heterologous assay using macrophages from healthy donors as effector cells as well as in an autologous setting supporting the clinical development of ISB1442 in both diseases [154].

The published results of clinical trials of bispecific antibodies are summarized in Table 1.

All the bispecific antibodies are still in early clinical trials and many questions remain to be answered. Available clinical data indicate an acceptable safety profile, suggesting that bispecific antibodies will soon enter current clinical practice. Despite the limited use, some weaknesses have been already identified. First, the ideal “format” of bispecific antibodies is still not defined; small molecules have a short half-life and require continuous or close infusion and frequent dose adjustments. Construct modifications increasing molecular weight may prolong half-life by preventing a rapid renal clearance, thus permitting delayed infusions, but may increase off-target toxicity. Second, antibodies themselves can favor the maintenance of a permissive microenvironment by inducing the upregulation of inhibitory costimulatory molecules, such as PD1/PD-L1, and by diminishing T-cell activation. In this instance, the concomitant administration of anti-PD1 antibodies could help, specifically in those leukemia subtypes known to have PD1 or PD-L1 upregulation. Finally, the best target on leukemic cells is far from being identified. Few antigens restricted to LSCs are known, and their expression (yes or no) and intensity can be variable and may change under chemotherapeutic pressure or at disease relapse. It must be underlined that the currently available information has been obtained in relapsed and often heavily pretreated subjects.

## 4. T- and NK-Cell Therapies

CAR-T cells are engineered autologous peripheral T cells equipped with a synthetic target-antigen receptor (CAR), able to expand after transfusion in a target-antigen-dependent matter and designed to persist after infusion, eventually generating a long-term anti-leukemic memory. Binding CAR to the target antigen initiates intracellular signaling, leading to TCR-independent T-cell activation. CAR’s structure consists of four main components, each of which can affect CAR-T functionality: (1) an extracellular target-antigen-binding domain, (2) a hinge region, (3) a transmembrane domain, and (4) one or more intracellular signaling domains. Extracellular domain structure is crucial not only for antigen recognition but also for the affinity and specificity of binding. Too low or too high affinity may result in the activation-induced death of CAR-T cells and trigger toxicities. The hinge region confers flexibility, allowing the antigen-binding domain to access the target epitope and form immunological synapses. The most-used hinge regions are derived from aminoacidic sequences from CD8, CD28, IgG1, and IgG4. The transmembrane domain influences CAR expression level and stability. Most transmembrane domains are derived from natural proteins including CD3ζ, CD4, CD8α, and CD28. A CD3ζ transmembrane domain probably facilitates T-cell activation, mediates CAR dimerization, and incorporates endogenous TCRs [155]. However, a CD3ζ domain is less stable than the CD28 one. The hinge region and transmembrane domain influence CAR-T cell cytokine production and activation-induced cell death (AICD), while intracellular domains influence the activation magnitude and persistence of CAR-T cells. The first generation of CAR-T cells has a single signal domain (CD3ζ), while the newer generations have additional costimulatory structures, such as CD28 or CD137 and 4–1BB for 2nd generation CAR-T and CD28plus, CD137, or CD134 (OX40) for the 3rd generation CAR-T [156]. The highly suppressive microenvironment and the fact that target antigens are also frequently expressed in normal hematopoietic counterparts represent a big challenge for CAR-T cell therapy. Nonetheless, the preliminary results of CAR-T cell therapy in AML demonstrated positive results, and more than 20 trials are currently ongoing. A summary of principal active trials is summarized in Table 2.

Target antigens under investigation in CAR-T cell therapy trials are CD33, CD123, CLL-1, CD38, NKG2D, and CD7.

CD33 is expressed in almost all AML subtypes. A Chinese clinical trial investigating the feasibility of anti-CD33 CAR-T therapy suggests a potential benefit but reports severe side effects (fever, pancytopenia CRS) [157]. Anti-CD123 CAR-T cells showed potent anti-leukemic activity in vitro, but difficulties in differentiating CD123 expression on leukemic cells and normal hematopoietic cells limit their utilization, and new strategies are needed to protect normal cells. CD44v6 is an adhesion protein supposed to favor leukemogenesis and contribute to the LSC phenotype. Anti-CD44v CAR-T cells have shown potent in vitro anti-leukemic effects, sparing normal HSCs. The feasibility and safety of CD44v6 CAR-T cells are under investigation in a phase I ongoing trial (NCT040097301). Even CLL-1 has been proposed as a target antigen for CAR-T cell therapy: in vitro studies and mice models showed encouraging results, with good distinction between normal and leukemic cells [158]. With these premises, a phase I trial using CD33-CLL1 dual CAR-T cells has been initiated (NCT03795779). The preliminary results presented at the 2020 European Hematology Association Annual Meeting reported MRD negativity in 7/9 treated patients [159]. The clinical therapeutic safety of CD38-targeted CAR-T cells was investigated by Cui and coll. in AML patients relapsing after allogeneic HCT (NCT04351022). Four weeks after infusion, 4/6 patients achieved CR/CRi with manageable side effects, and LFS was 7.9 months [160]. Ultra-CAR-T, simultaneously expressing CD33 and IL-15, was used in R/R AML and high-risk MDS patients in a phase I dose-finding, trial (NCT03927261): ORR was 50%, with good safety and tolerability [161]. Preclinical studies evaluating the double targeting of FLT3 with CAR-T cells and the FLT3 inhibitor gilteritinib seem to suggest potent leukemia clearance and normal HSC protection [162]. Appelbaum and coll. explored the use of controlled anti-CD33 CAR-T cells (SC-DARIC33), genetically modified to express a Dimerizing Agent Regulated Immunoreceptor Complex (DARIC) controlled by post-infusion intermittent doses of rapamycin in a phase I trial in pediatric and young adult patients with R/R CD33+ AML (NCT05105152). Finally, NK CAR cells targeting different AML cell antigens (CD70, CD33, and FLT3, CLL-1) exhibit potent anti-leukemic effects “in vitro” for their powerful cytotoxic effects with limited off-target side effects and unique recognition mechanism [163,164,165] and are at present in preclinical evaluation.

Table 3 summarizes the data on T-NK-cell therapy presented at the 64th ASH Annual Meeting.

CAR-T cell therapies are expected to revolutionize the treatment of hematologic malignancies; however, beyond the enthusiasm for being the first truly innovative cellular therapy since allogeneic transplantation, many obstacles remain to be overcome. In AML, the ideal target antigen should be specific for LSCs, constantly expressed through all disease phases, and easy to monitor. At the same time, CAR-T cell production requires refinements to improve target antigen recognition and assure a binding strength sufficient to activate T cells without causing activation-induced cell death, limiting off-target toxicity and prolonging CAR-T cell persistence.

## 5. Conclusions

The enormous advances made in the past years regarding molecular and cellular mechanisms leading to immune dysfunction and leukemia tolerance, the encouraging results obtained in lymphoproliferative diseases and in solid tumors, and the advances in molecular analysis and sequencing technologies have generated great expectations regarding the possibility to manipulate the immune system and BM niches in AML. Despite interesting preliminary data, we are far from having a “magic bullet” to eradicate leukemia cells. Effective targets on AML cells must be identified, but this effort is complicated by the fact that LSCs and HSCs share various antigens, making immune therapy toxicities often unacceptable, and by a mutation rate that in AMLs is 40-fold less compared to solid tumors, significantly reducing the chance of appearance of neo-antigens. This could be counterbalanced by much easier accessibility to AML cells and by their susceptibility to killing.

The possibility to target the BM niche, thus reversing the immunosuppressive microenvironment, is a promising approach. Combined therapies need to be studied to harness immune function, taking advantage of the peculiar activities of conventional drugs, such as cytarabine, which is known to increase the expression of MHC class II and CD80 costimulatory molecules and to downregulate PD-L1, or hypomethylating agents that increase the expression of many immune-related genes, or multi-tyrosine kinase inhibitors that can induce the release of cytokines potentiating T and NK response.

## Figures and Tables

**Figure 1 cancers-15-00253-f001:**
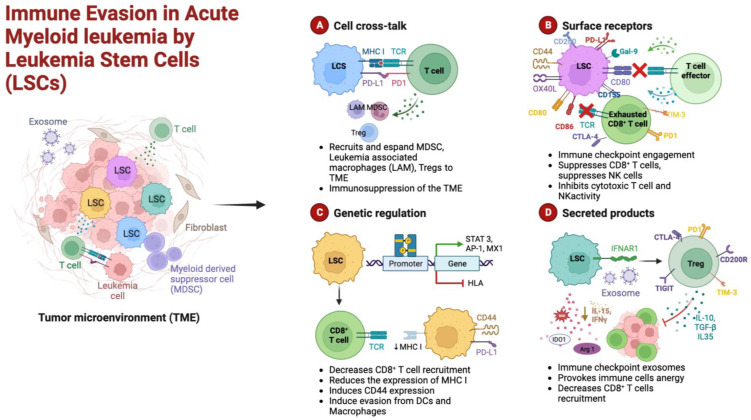
Schematic representation of the cellular mechanisms of anti-tumor surveillance escape in AML. (**A**) Cellular cross-talks: leukemic stem cells (LSCs) and T cells direct interaction, through major histocompatibility complex (MHC) and over-expressed immune checkpoint ligands (PD-L1), causes functional defects in anti-tumor immunity. LSCs induce the expansion of immunosuppressive cells such as regulatory T cells (Tregs), myeloid-derived suppressor cells (MDSCs), and leukemia-associated macrophages (LAMs). (**B**) Surface receptors: LSCs can hamper both T- and NK-cell effector functions by aberrantly overexpressing inhibitory ligands such as CD155, CD80, PD-L1, and galectin-9 (Gal-9). (**C**) Genetic regulation: the activation of immune checkpoint pathways, such as PD-1 and cytotoxic T-lymphocyte-associated protein 4 (CTLA-4), and the expression of CD44, a cell-adhesion molecule that has been shown to be involved in the promotion of resistance to drug-induced apoptosis, prevent the appropriate anti-tumor immunity. Moreover, epigenetic mechanisms may result in the downregulation of MHC expression in AML cells, leading to immune escape and relapse in AML. (**D**) Secreted products: LSCs release inhibitory mediators, either as soluble molecules or as a component of exosomes; furthermore, induced immune cell (Tregs, CD8+ T cells) anergy lowers the levels of inhibitory cytokines. Among these factors that alter the bone marrow microenvironment milieu, a relevant role is played by indoleamine 2,3-dioxygenase (IDO), interferon-gamma (IFN-γ), transforming growth factor beta (TGF-β), arginase 1 (Arg 1), and interleukin 10 and 15 (IL-10 and IL-15). Figure created in Biorender.com.

**Figure 2 cancers-15-00253-f002:**
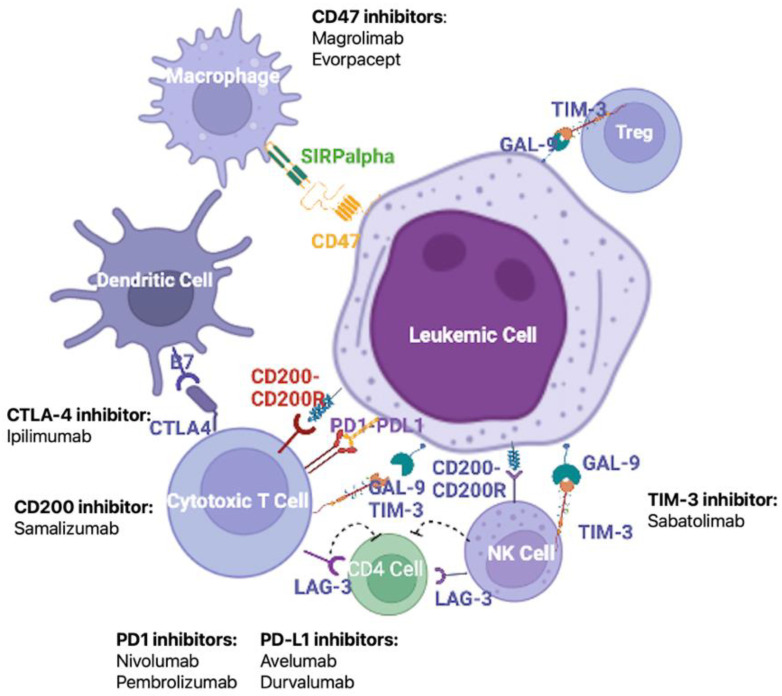
Major mechanisms of the maintenance of immune tolerance and potential therapeutic targets. Given the pivotal role of LSC-induced immune escape in the development and relapse of AML, an important treatment strategy to restore the function of anti-tumor immune cells is the blockade of immune checkpoints by monoclonal antibodies. Among the most promising targets are CD47, cytotoxic T-lymphocyte-associated protein 4 (CTLA-4), CD200, the PD1/PD-L1 axis, and T-cell immunoglobulin and mucin domain-containing protein (Tim-3). Figure created in Biorender.com.

**Table 1 cancers-15-00253-t001:** Published results of clinical trials of bispecific antibodies in AML.

	Target	Agent	Phase	Disease	Outcome	NCT ID
BiTE	CD33 × CD3	AMG330	I	R/R AML	CR+CRi:19%	NCT02520427
					CRS:67%	
	CD33 × CD3	AMV564	I	R/R AML	CR+CRi:66.7%	NCT03144245
					CRS:5.7%	
	CD33 × CD3HLE	AMG673	I	R/R AML	Blast reduction:44%	NCT03224819
					CRS:50%	
	CD123 × CD3	APVO436	I	R/R AML	Blast reduction: 7%	NCT03647800
					CRS:18%	
	CD123 × CD3	XmAb14045	I	R/R AML	CR+CRi:14%	NCT02730312
		(Vibecotamab)			CRS:59%	
DART	CD123 × CD3	Flotetuzumab	I/II	R/R AML	CR/CRi:27%	NCT02152956
					CRS13%	
TRiKE	CD33 × CD16 × IL15	1-GBT-3550	I	PIF R/R AML	SD:50%; POD:25%	NCT03214666
					No toxicity	

BiTE: bispecific T-cell engager; DART: Dual-affinity Re-Targeting Molecules (BiTE); HLE: half-life extended; TRiKE: tri-specific killer engager; R/R: relapsed/refractory; CR: complete remission; CRi: complete remission with incomplete hematologic recovery; SD: stable disease: POD: progression of disease; PIF: primary induction failure; CRS: cytokine-release syndrome.

**Table 2 cancers-15-00253-t002:** Summary of preliminary results of CAR T/CAR NK cell therapy for AML.

Product	Target Antigen	Phase	Disease State	Clinical Outcome	NCT ID
CAR T (2nd gen)	Lewis-Y Ag	I	R/R	Transient blast decrease 1/4pts	NCT01716364
	CD33	I	R/R	Marked blast decrease 1pt	NCT02799680
CAR T	NKG2D	I	R/R	CRh+CRi 3/7pts	NCT02203825
Compound CAR T	CLL-1 and CD33	I	R/R	MRD neg 1 pt	NCT03795779
CAR-NK	CD33	I	R/R	Decrease in MRD 1/3pts	NCT02944162

R/R: relapsed/refractory; MRD: minimal residual disease; CRh: hematologic complete remission; CRi: complete remission without complete hematologic recovery.

**Table 3 cancers-15-00253-t003:** Preclinical and clinical studies of CAR-T and CAR-NK cells for acute myeloid leukemia at the 2022 ASH annual meeting.

Author (REF)	Albinger [166]	Ehninger [167]	Sallman [168]	Naik [169]	Kloos [170]	Sallman [161]
Study type	Preclinical	Clinical (phase I, dose-escalation)	Clinical (phase I, dose-escalation)	Clinical	Preclinical	Clinical
Target	CD33	CD123	CD123	CD123	CD7/CD33	CD33
Cell source	NK	UniCAR-T	Anti. CD123 allogeneic CAR-T	CAR-T	CAR-T	Ultra CAR-T
Disease	AML	R/R AML, CD123+	R/R AML, CD123+	R/R pediatric AML, CD123+	AML cells	R/RAMLs and MDSs
Innovation	NKG2A-KO	CD28 costimulatory domain	TRAC and CD52 gene disruption to minimize GVHD	Bridge to allo-HCT	Double target on leukemia cells	Membrane-bound Il-15
Setting	In vitro and in vivo (mice)	In vivo (14 patients)	In vivo (16 patients)	In vivo (12 patients)	In vivo (mice)	In vivo (24 patients)
Results	In vitro AML increased cytotoxicity In vivo AML cells and leukemia-initiating cell elimination	Good safety and tolerability CRS grade 1–2 (12 patients), CRES (1 patient), blast count reduction (10 patients), CRi (2 patients), flow cytometric MRD negativity (1 patient)	Good safety and tolerability CRS 15/16 (≥3 3 patients) Evidence of UCART123 activity 4/16 SD, 2 patients; blast cell reduction, 1 patient; MRD-negative CR, 1 patient)	Good safety and tolerability No grade 2 CRS or CRES No response (2 patients); reduction in blast cells (1 patient); CR (1 patient)	Depletion of AML cells to 2.6–2.9% Prolonged survival	Good safety and tolerability Grade 3 CRS (1 patient) Dose-dependent expansion of Ultra CAR-T; durable persistence 30% ORR: 1 CRi; 1 CRh; 1 PR No response in MDSs

NKG2A: natural killer group 2A; KO: knockout; TRAC: T-cell receptor alpha constant; GVHD: graft versus host disease; SD: stable disease; MRD: minimal residual disease; CR: complete remission; CRi: complete remission with incomplete hematologic recovery; CRS: cytokine-release syndrome; CRES: CAR-T related encephalopathy syndrome; allo-HCT: allogeneic hematopoietic cells transplantation; PR: partial response.

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
