# Peer review of "Present and Future Role of Immune Targets in Acute Myeloid Leukemia"

_cancers, 2022, doi:10.3390/cancers15010253_

Round 1
Reviewer 1 Report
Thank you for the opportunity to review the article on ‘Immune targets in acute myeloid leukemia: where we are and where we are going’ by Damiani et al. I commend the authors on this effort in this review. Here are a few suggestions for improvement:
1. The standard abbreviation for Leukemic Stem Cells is LSCs. You have this correct in some forms, but most is ‘LCS’ which is inconsistent: lines 62, 66, 68, 75, 76, 80, 82, 104, 221, 636. I would recommend using one or the other abbreviation.
2. For paragraph structure, "Also" as seen in line 102 is not a strong thesis sentence. I would remove this sentence altogether as the point is made starting on Line 103.
3. Under section 3.1, line 210: this line can be omitted as it does not add anything to the paragraph and is not a strong introduction to this section.
4. Line 324: the authors mention that aberrant expression of CTLA-4 has negative impact on disease outcome, I would recommend providing a number to support this statement and add a percentage of cases that have aberrant CTLA-4. This would benefit the discussion of PDL-1, PDL-2.
5. Line 538, the authors have mentioned E:T ratio, however I recommend adding some discussion on what that necessarily means in the context of the sentence.
6. Overall, some of the information/ thoughts presented would do better if expanded into separate paragraphs. For instance, Magrolimab starting Line 418 would better be served as the start of a paragraph.
7. Please rewrite lines 544-545 as it does not convey the point coming across that AMG427 causes not only cytotoxicity, but increases the production of PD1 expression which can act as an immune escape mechanism without concurrent administration of anti-PD1 therapy.
Author Response
Thank you for the opportunity to review the article on ‘Immune targets in acute myeloid leukemia: where we are and where we are going’ by Damiani et al. I commend the authors on this effort in this review. Here are a few suggestions for improvement:
We thank the reviewer for his/her positive comments and the appreciation of our work.
- The standard abbreviation for Leukemic Stem Cells is LSCs. You have this correct in some forms, but most is ‘LCS’ which is inconsistent: lines 62, 66, 68, 75, 76, 80, 82, 104, 221, 636. I would recommend using one or the other abbreviation.
We uniformed the text using the acronym LSCs throughout the manuscript.
- For paragraph structure, "Also" as seen in line 102 is not a strong thesis sentence. I would remove this sentence altogether as the point is made starting on Line 103.
We removed the sentence.
- Under section 3.1, line 210: this line can be omitted as it does not add anything to the paragraph and is not a strong introduction to this section.
We removed the sentence.
- Line 324: the authors mention that aberrant expression of CTLA-4 has negative impact on disease outcome, I would recommend providing a number to support this statement and add a percentage of cases that have aberrant CTLA-4. This would benefit the discussion of PDL-1, PDL-2.
We have reported the results of a work from C. Chen et al published as a scientific letter in Journal of Hematology & Oncology 2020; unfortunately, the manuscript does not provide details on aberrant CTLA-4 expression.
- Line 538, the authors have mentioned E:T ratio, however I recommend adding some discussion on what that necessarily means in the context of the sentence.
The sentence was modified, adding the term “low” before “E:T ratio”.
- Overall, some of the information/ thoughts presented would do better if expanded into separate paragraphs. For instance, Magrolimab starting Line 418 would better be served as the start of a paragraph.
We modified the paragraphs’ structure as suggested.
- Please rewrite lines 544-545 as it does not convey the point coming across that AMG427 causes not only cytotoxicity but increases the production of PD1 expression which can act as an immune escape mechanism without concurrent administration of anti-PD1 therapy.
We have reformulated the sentence to correctly convey the message.

Reviewer 2 Report
This is an overall nicely written review summarizing the available data with several immune-based therapeutic modalities such as monoclonal antibodies, T cell engagers, adoptive T-cell therapy, adoptive-NK therapy, checkpoint and macrophage checkpoint blockade.
I have minor comments for the authors:
1- I suggest changing/rewriting the title so it does not seem to be confused with another paper with similar scopes published in 2021 (https://doi.org/10.3389/fonc.2021.656218)
2- The ongoing trials could be summarized in a table stating the receptors, population, endpoints and NCT.gov registration
3- The figures captions should be expanded and explained
4- The paper would currently stand out in the literature if it adds the findings just presented at ASH. I suggest the authors add this information to each section
Author Response
This is an overall nicely written review summarizing the available data with several immune-based therapeutic modalities such as monoclonal antibodies, T cell engagers, adoptive T-cell therapy, adoptive-NK therapy, checkpoint and macrophage checkpoint blockade.
We thank the reviewer for his/her constructive comments.
I have minor comments for the authors:
- I suggest changing/rewriting the title, so it does not seem to be confused with another paper with similar scopes published in 2021 (https://doi.org/10.3389/fonc.2021.656218)
We thank the reviewer for having highlighted this potentially confounding factor: we thus changed the title to “Present and future role of immune targets in acute myeloid leukemia”.
2- The ongoing trials could be summarized in a table stating the receptors, population, endpoints and NCT.gov registration
As there are dozens of ongoing studies, most of which with little or no published results, we felt that a mere list of trials would add little information. However, we reported in the text the status of 48 trials of CAR-T therapy in AML registered in NCT.gov.
3- The figures captions should be expanded and explained
Figure legends have been expanded.
4- The paper would currently stand out in the literature if it adds the findings just presented at ASH. I suggest the authors add this information to each section
We added in a Table the results of CAR-T trials presented at ASH 2022 Annual Meeting; more, we briefly reported in each section the most relevant updated presented at the same Meeting on checkpoint inhibitors and other immune therapies for AML.
